# MEAN REPLACEMENT PRUNING

## ABSTRACT

Pruning units in a deep network can help speed up inference and training as well as reduce the size of the model. We show that *bias propagation* is a pruning technique which consistently outperforms the common approach of merely removing units, regardless of the architecture and the dataset. We also show how a simple adaptation of an existing scoring function allows us to select the best units to prune. Finally, we show that the units selected by the best performing scoring functions are somewhat consistent over the course of training, implying the dead parts of the network appear during the early stages of training.

## 1 INTRODUCTION

Pruning is a successful method for reducing the size of a trained neural network and accelerating inference. Pruning consists of deleting the parts of the network whose removal least affects the network performance. Many pruning methods proposed in the literature differ in computational cost and in effectiveness in ways that are hard to assess.

In an interesting recent work, Frankle & Carbin (2018) argue for the so called "*winning ticket*" hypothesis. More precisely, they train a large network after saving the random initial value of each parameter. After training, they prune the large network to produce a smaller network with one fifth of the weights. Setting its weights to their saved initial values and retraining achieves a performance close to that of the large trained network with a much reduced computational cost. This result opens up a new frontier for pruning methods, where they are used to detect useless units early in the training and therefore accelerating the inference.

This contribution studies the effect of pruning methods throughout the training process. We also present *mean replacement*, a unit pruning method that extends the idea of bias propagation introduced in (Ye et al., 2018) to the non-constrained training setting. The main observations of our work can then be summarized as follows:

- Regardless of the scoring function used, bias propagation reduces the pruning penalty for networks without batch normalization.

- Fine-tuning the pruned network with additional training iterations reduces the bias propagation advantage but not very quickly.

- Absolute valued approximation of the pruning penalty provides superior performance over the normal first order approximation. This finding confirms the observations made by Molchanov et al. (2016).

- Units that are selected by the best performing scoring function seem to come from a small subset of units. This finding confirms Frankle & Carbin (2018)'s comments on the lottery ticket and Evci (2018)'s claims about dead units.

The rest of the paper is organized as follows. After reviewing the related work in Section 2. we define our pruning methods and scoring functions in Section 3. Section 4 provides an empirical evaluation comparing various combinations of scoring functions and methods under varying pruning fractions, datasets, and models. We briefly provide some concluding remarks and discuss future work in Section 5.

## 2 RELATED WORK

One simple and common technique to select what parts of a network to prune is to select small magnitude parameters, similar to the technique of weight decay (Hanson & Pratt, 1989). LeCun et al. (1990) and Hassibi & Stork (1993) proposed using second-order saliency measures to prune trained networks with zero gradient. With the era of deep networks, redundancy in trained networks became even more obvious and various works tackled this problem aiming to reduce the size of the network. Han et al. (2015) applied magnitude based parameter pruning on deep networks and reported around 30x compression combining various methods like weight quantization and Huffman coding. Zhu & Gupta (2017) perform pruning during training and report similar compression rates. Achterhold et al. (2018) focus on pruning Bayesian neural networks.

Evci (2018) claims that the removed parameters tend to gather around specific units, so directly pruning full units might prove to be an efficient strategy. Further, pruning units also comes with direct gains in terms of storage and speed since it meshes well with dense representations (Luo et al., 2017; Wen et al., 2016). Wen et al. (2016) prune entire channels using the group lasso penalty. Hu et al. (2016) observe high percentage of zero activations in deep networks with ReLU units and propose Average Percentage of Zeros as a new saliency function. The work of Molchanov et al. (2016) focuses on iterative pruning with single unit removals in the context of transfer learning. They compare various scoring functions and propose the absolute-valued Taylor approximation as the best performing one. However, one practical drawback of their investigation is that they prune one unit at a time.

Once units have been selected for removal, there is still the question of how to minimize the impact of the deletion. Most works, (Molchanov et al., 2016) for instance, focus on mere removal of the units followed by retraining the network. While retraining the full network after pruning can greatly minimize the loss in accuracy induced by the deletion, it can be computationally expensive.

Rather than recovering from the damage post-pruning, another line of research focuses on preemptively mitigating the effects. Ye et al. (2018) propose penalizing the variance of the activations for networks using batch normalization. They then propose replacing units with low variance with constant values using *bias propagation*. Our work extends this idea of bias propagation to various other pruning methods. Morcos et al. (2018) suggest ablating units (which mimics removal) by replacing them with their mean activation; however, the authors report this yields inferior performance compared to simply removing the units. Luo et al. (2017) propose using the l2-norm of unit activations to iteratively prune convolutional layers for VGG-16 and ResNet. They also propose performing updates on the outgoing weights that minimize the reconstruction loss on the next layer. Their method relies on a matrix inversion rendering it impractical for large networks.

## 3 UNIT PRUNING AND MEAN REPLACEMENT IDEA

Even with a careful unit selection, pruning can significantly damage the performance of the network. Fine-tuning the damaged network with retraining iterations may or may not recover the full performance. Thus our goal is to minimize the damage as much as possible at pruning time. In this section we introduce mean replacement, a simple pruning method that significantly reduces the loss incurred by the ablation.

### 3.1 PRUNING UNITS AND NOTATION

For a given dataset $\mathcal{D}$ with samples $(\boldsymbol{x}^{(i)}, y^{(i)})$, output $f(\boldsymbol{x}^{(i)}; \boldsymbol{w})$ using parameters $\boldsymbol{w}$, and loss function $l(f(\boldsymbol{x}^{(i)}; \boldsymbol{w}), y^{(i)}; \boldsymbol{w})$, the loss of the optimization problem is

$$L(\boldsymbol{w}) = \sum_{i \in \mathcal{D}} l(f(\boldsymbol{x}^{(i)}; \boldsymbol{w}), y^{(i)}) .$$

Pruning a network is often defined as setting some of its parameters to 0. Given $\boldsymbol{w}$, our goal then consists of finding the mask $\boldsymbol{m} \in \{0, 1\}^d$ that minimizes

$$L(\boldsymbol{w} \odot \boldsymbol{m}) = \sum_{i \in \mathcal{D}} l(f(\boldsymbol{x}^{(i)}; \boldsymbol{w} \odot \boldsymbol{m}), y^{(i)}) .$$

The mask $\boldsymbol{m}$ must respect some constraints. First, even though the mask is defined at the parameter level, i.e. it has as many components as the number of parameters in the model, we are pruning units. Hence, all parameters corresponding to the same unit must have the same value in the mask. Second, we are interested in pruning only a limited number of units, so the number of elements set to 0 in $\boldsymbol{m}$ is constrained. Finally, we might also want to enforce the number of units pruned at each layer, or simply prevent the pruning at some layers. Denoting as $\mathbb{M}$ the set of all masks satisfying these constraints, the optimal pruning is given by

$$\boldsymbol{m}^* = \arg\min_{\boldsymbol{m} \in \mathbb{M}} \sum_{i \in \mathcal{D}} l(f(\boldsymbol{x}^{(i)}; \boldsymbol{w} \odot \boldsymbol{m}), y^{(i)}) \ . \tag{1}$$

For the remainder of the paper, we shall assume that the number of units to remove is set for each layer independently. This is without loss of generality and will allow us to focus on a single layer, greatly simplifying the presentation.

The complexity of solving Eq. 1 increases exponentially with the number of units to prune. Therefore, in practice, people rank all units using a per-unit scoring function $s(\boldsymbol{w}; \mathcal{D})$. Several examples of such scoring functions will be discussed in Section 3.4. Units with the lowest score are then pruned. This approach implicitly assumes that the scores of individual units are independent of each other. In other words, pruning one unit is assumed to not affect the score of any other unit. Thus, a good scoring function needs to have a small inter-unit correlation. Once a scoring function $s(\boldsymbol{w}; \mathcal{D})$ has been chosen, we can define $\boldsymbol{m}$ through its elements $m_i$:

$$m_i = \begin{cases} 0 & \text{if } i \text{ belongs to a unit } u \text{ where } u \in B(s(\boldsymbol{w}; \mathcal{D}), k) \\ 1 & \text{otherwise} \ . \end{cases} \tag{2}$$

where $B(\boldsymbol{x}, k)$ is the set of $k$ elements of $\boldsymbol{x}$ with the lowest value. In other words, we will set to 0 all the parameters belonging to units whose score is one of the $k$ lowest, $k$ being the number of units we wish to remove in that layer.

Pruning a fraction of the units in a particular layer can have a big impact on the network and induce a large loss penalty, that we call the *pruning penalty*:

$$PruningPenalty = L(\boldsymbol{w} \odot \boldsymbol{m}) - L(\boldsymbol{w}) \tag{3}$$

Retraining the network might reduce the pruning penalty at the cost of additional computation. We shall now see how adjusting the biases of the following layer can reduce the pruning penalty with low computational overhead. In order to show this, we need to depart from our earlier definition of pruning as consisting of zeroing a subset of the weights.

## 3.2 MEAN REPLACEMENT

We intend to remove $k$ units from a layer of the network in a manner that has a reasonable impact on the network performance. This is often done by replacing these units with zeroes. However, zero is an arbitrary choice and any constant would work. This constant would be "propagated" by multiplying it with the outgoing weights of the layer above, which is equivalent to updating the bias of that layer with the resulting sum. *Mean replacement* consists of replacing the output of pruned units by a constant that is equal to the mean of the unit outputs collected on the training samples before pruning. A theoretical justification for that choice will be presented in Section 3.3.

We will first focus on the removal of a single unit. In a fully connected network, each unit is associated with a single activation. However, in a convolutional layer, a unit is associated with a set of outputs, one per location. In that case, each of these output will be replaced with the same constant.

Let $a(x, p)$ represent the unit output for training example $x$ at location $p \in \mathcal{P}$. Let us randomly choose a subset[1] $\mathcal{D}_s \subset \mathcal{D}$ of examples from the training set. We first compute the mean unit output

$$\bar{a} = \frac{1}{|\mathcal{D}_s|} \sum_{x \in \mathcal{D}_s} \frac{1}{|\mathcal{P}|} \sum_{p \in \mathcal{P}} a(x, p) \ .$$

---

[1]Usually smaller than the full training set, but big enough to get a good approximation.

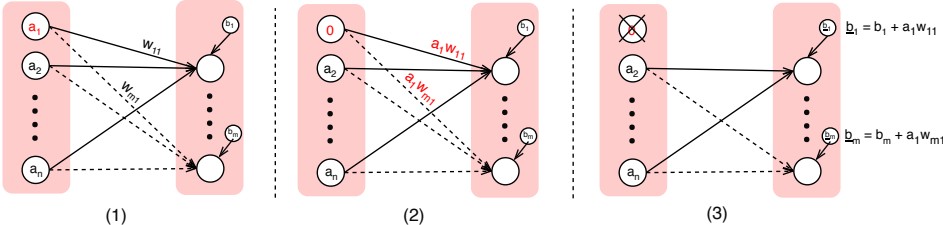

Figure 1: Mean Replacement illustrated in three steps. In step (1) the units to be pruned are selected (highlighted in red). In step (2) mean activations are multiplied with outgoing weights. In step (3) the product is added to the bias of corresponding units.

Mean replacement consists in replacing the pruned unit by the constant $\bar{a}$. This can be implemented by removing the pruned unit in the normal way —which amounts to replace its output by a zero— and folding the constant $\bar{a}$ into the bias parameter of the downstream units.

$$\boldsymbol{b} \leftarrow \boldsymbol{b} + \bar{a}\boldsymbol{w} \,, \tag{4}$$

where $\boldsymbol{b}$ represents the vector of the biases of the downstream units and $\boldsymbol{w}$ represents the outgoing weights of the pruned unit, that is the weights that were connecting the pruned unit to each of its downstream units before the pruning operation. This process is illustrated in Figure 1.

We now justify our choice of constant by showing that, in the context of a quadratic loss, mean replacement is the optimal strategy.

### 3.3 OPTIMAL BIAS UPDATE FOR LINEAR REGRESSION

Let us consider the linear regression setting with $K$ samples $(\boldsymbol{x}^{(i)}, y^{(i)}|i \in [1, K])$ and parameters $\theta$ and $b$ where $h(x^{(i)}) = \theta^T \boldsymbol{x}^{(i)} + b$. Let us write down the optimal bias for the mean square loss $L = \frac{1}{2K}\sum_{i=1}^{K}(h(x^{(i)}) - y^{(i)})^2$: $b^* = \frac{1}{K}\sum_{i=1}^{K}(y^{(i)} - \theta^T \boldsymbol{x}^{(i)})$.

Let us consider the case where we prune the input dimension $j$ and denote the pruned samples with $x_{-j}^{(i)}$. Then we would have the objective $L_{-j} = \frac{1}{2K}\sum_{i=1}^{K}(h(x_{-j}^{(i)}) - y^{(i)})^2$ where $h(x_{-j}^{(i)}) = h(\boldsymbol{x}^{(i)}) - \theta_j x_j^{(i)}$. The optimal bias value for this new setting is $b_{-j}^* = \frac{1}{K}\sum_{i=1}^{K}(y^{(i)} - \theta^T \boldsymbol{x}^{(i)} + \theta_j x_j^{(i)})$.

The difference between these two optimal bias values would give us the optimal update value for the bias of the next layer after pruning, which is indeed the mean values of the pruned dimension.

$$c = b_{-j}^* - b^* = \frac{1}{K}\sum_{i=1}^{K}\theta_j x_j^{(i)}$$

One can easily show that the optimal value is the sum of propagated inputs, if we prune more then one input features.

Although motivating through linear regression might not seem relevant in the deep learning case, the activations $\boldsymbol{a}^l$ at layer $l$ can be viewed as the input of the linear regression. Each channel of the the linear function $h(\boldsymbol{a}^l)$ can be thought as a separate linear regression. Using this observation, we can minimize the l2-norm between activations before pruning and activations after pruning, namely $||h(\boldsymbol{a}^l) - h(\bar{\boldsymbol{a}}^l||)$, fixing the weights. Luo et al. (2017) take a very similar approach motivating their pruning method. They find the optimal update without fixing the weights, requiring matrix inversion of a matrix size $|\mathcal{D}_s|$. What is the optimal update for the bias in next layer? As in the case of linear regression, we can show that the optimal update is the Mean Replacement.

### 3.4 SCORING FUNCTIONS FOR UNIT PRUNING

Most practical pruning methods use scoring functions that assign scores to individual units. These scoring functions attempt to assign a score to each unit such that units with small scores have the

smallest loss degradation ($\Delta L$) when pruned separately. In practice, however, scoring functions that were designed for single unit removal are used to prune $k$ units at once. This is valid as long as there is no cross-correlation between the scores of individual units, but this is often not the case: removing one unit usually changes the scoring distribution and possibly invalidates the previous ordering among units.

Since we aim to do unit pruning with minimal overhead, the complexity of all the scoring functions included in our experiments are linear with size of the layer or the cardinality of $\mathcal{D}_s$. Throughout our experiments we compare 6 scoring functions as summarized in (Table 1) and the features of them explained below.

**Type.** There are 4 different types of scoring functions used in our experiments. Our baseline is the *random* which samples scores uniformly from the range [0,1]. *norm* is the l2-norm of the unit. One common phenomenon in training neural networks with a softmax is that the norm of the parameters tend to increase over training (Raghu et al., 2017). We can thus expect units that are not contributing much to the learning process to have smaller norms. *abs_taylor* and *taylor* are the Taylor approximation having the form $\mathbb{E}_{i \sim \mathcal{D}_s} |\nabla_a L(a) \odot \Delta a|$ and $\mathbb{E}_{i \sim \mathcal{D}_s} (\nabla_a L(a) \odot \Delta a)$ respectively. In our experiments we use a subset $\mathcal{D}_s$ sampled from the training set of size 1000 (Cifar-10) and 10000 (Imagenet). *taylor* is the correct first order approximations for the change in the loss. However, in practice, to our knowledge, they are not used without the absolute values. In our experiments we confirm that they perform significantly worse compared to the other scoring functions. We discuss the possible reasons and our observations in Section 4.3.

**Approximated Penalty.** Indicates the value that is being approximated. *mean replacement* indicates the pruning penalty if mean replacement is used, whereas *removal* indicates the penalty with regular pruning where the pruned units are just set to zero.

**Mean Replaced?.** This column indicates whether the pruning method itself has the bias propagation step.

## 4 EMPIRICAL EVALUATION

We use the following experimental approach to compare various pruning strategies. At various points during the network training, we make a copy of the network, prune a predefined fraction of its units using the chosen criterion, and measure the pruning penalty by comparing the losses measured before and after pruning. We then resume the training process using the original copy of the network (prior to pruning). We repeat this experiment for different convolutional networks with different sizes and depths on Cifar-10 (Krizhevsky, 2009) and Imagenet-2012 (Russakovsky et al., 2015) initialized using various random seeds. In all of our experiments we calculate the pruning penalty using subsets of sizes 1000(Cifar-10) and 10000(Imagenet) sampled from the training set. Appendix 6.2 details the full set of experiments.

Table 1: Scoring Functions Compared

| Type | Approximated Penalty | Mean Replaced? | Referenced in | Plot Tag |
|---|---|---|---|---|
| norm | - | ✗ | Luo et al. (2017) | norm |
| | | ✓ | ours | bp_norm |
| random | | ✗ | - | rand |
| | | ✓ | - | bp_rand |
| abs_taylor | mean replacement | ✗ | ours | abs_mrs |
| | | ✓ | ours | bp_abs_mrs |
| | removal | ✗ | Molchanov et al. (2016) | abs_rs |
| | | ✓ | ours | bp_abs_rs |
| taylor | mean replacement | ✗ | - | mrs |
| | | ✓ | - | bp_mrs |
| | removal | ✗ | - | rs |
| | | ✓ | - | bp_rs |

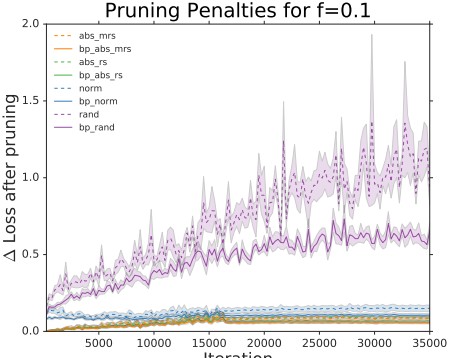
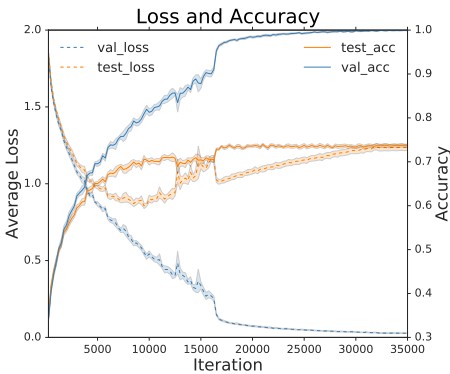

Figure 2: **(left)** Results from a single experiment, where MEDIUM_CONV (a 5 layer convolutional network) trained for 35k iterations with batch size 64. We calculate pruning penalties at every 250 iteration after pruning 10% of the units at each layer . All the measurements are made on a copied model and the mean values over 8 runs reported with 80% confidence intervals. **(right)** We plot the average loss on validation and test sets using the left axis and accuracies on the right.

## 4.1 MEAN REPLACEMENT REDUCES THE LOSS AFTER PRUNING

To assess the effectiveness of bias propagation across a wide variety of settings we trained various networks using the same learning rate schedule but different pruning fractions. We pruned various combinations of layers from pruning a single layer to all layers at once. A copy of the network was pruned every 250 steps during training, and we report the pruning penalty at these points. Figure 2 evaluates the performance of various pruning methods over the training of a five layer convolutional network and demonstrates that bias propagation reduces the pruning penalty for all pruning methods considered. Figure 3 aggregates all such measurements plotting (x,y) pairs from each scoring function at every time step, where x-axis denotes the pruning penalty without bias propagation used and y-axis denotes the penalty with the bias propagation. The cloud of points under the $y = x$ line shows that bias propagation decreases the pruning penalty in almost all cases despite the variety of settings the points are sampled from (different pruning fractions, layers pruned, models trained).

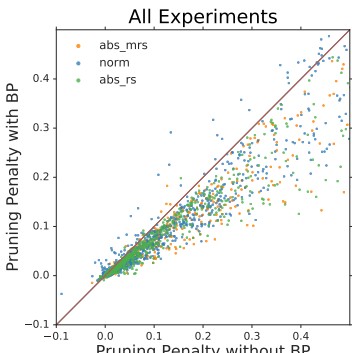
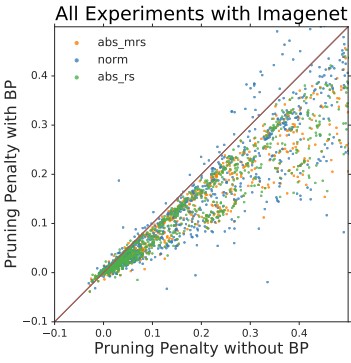

Figure 3: Scatter plots aggregating all measurements, where the two pruning penalties (with and without bias propagation) for the same scoring function are plotted in the opposite axes.**(left)** Experiments with Cifar-10 dataset: pruning penalties are calculated every 250 training iterations. **(right)** Experiments with Imagenet-2012 dataset: pruning penalties are calculated every 10000 training iterations.

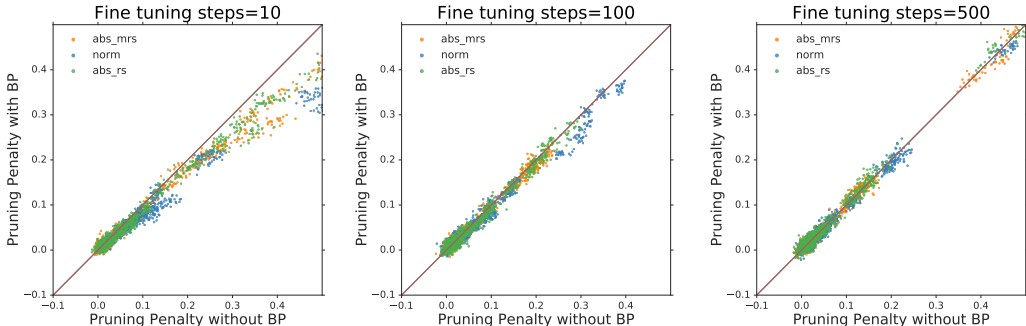

Figure 4: Pruning penalties after retraining the network with batch size 64 and learning rate 1e-3 for $N$ fine tuning steps. Pruning penalties are measured after fine tuning steps and aggregated in a scatter plot. The data used originate from pruning experiments on Cifar-10 made after 25000 training steps (second half of the training). **(left)** N=10. **(middle)** N=100. **(right)** N=500

## 4.2 MEAN REPLACEMENT REDUCES THE LOSS AFTER PRUNING AND RETRAINING

One could argue that training the pruned network could quickly compensate for the damage caused by zeroing the units without bias propagation. In other words, the networks pruned without mean replacement might end up learning the correct bias quickly through fine tuning, achieving the same loss as the network pruned with mean replacement after $N$ fine tuning steps. To assess this claim, we repeat our basic experiments but perform a specific number of retraining steps before measuring the post-pruning loss. In order to eliminate the unstable effects observed during the early stages of training, in this experiment we only consider the pruning-and-retraining penalties measured after at least 25,000 training iterations on the Cifar-10 dataset. Most of the networks we train have near zero losses by that time (see Figure 2 (**right**)). Figure 4 shows the scatter plots for 3 different values of fine tuning iterations. Although the effect of Mean Replacement diminishes when we increase the number of fine tuning steps, we can still see a difference after 500 fine tuning steps, which is almost one full epoch. This observation supports our claim that the immediate improvement on pruning penalty helps the future optimization.

In Appendix 6.5 we share the plots sampled from the other half of the results (networks pruned before the training step 25k). And finally in Appendix 6.6 we perform some iterative pruning experiments where we see the networks pruned with various methods converge almost to the same energy level when trained long enough.

## 4.3 FIRST ORDER APPROXIMATIONS OF THE LOSS PENALTY ARE UNRELIABLE

In this section we compare the performance of different scoring functions under our methodology. To summarize the results for all experiments without losing the distance information provided by a time series plot like the one in Figure 2-**(left)**, we use performance profiles (Dolan & Moré, 2001). We include measurements from all Cifar-10 experiments to generate the performance profiles for all the pruning methods considered in our work.

Let us denote measurement $j$ with tuples $\boldsymbol{d}^{(j)} = (d_1, ..., d_{12})$ where $d_i$ is the pruning penalty for $i$'th pruning method. Then for each such tuple we set the threshold to be $t_j = min(\boldsymbol{d}^{(j)}) * \tau + max(\boldsymbol{d}^{(j)}) * (1 - \tau)$ for each data point $j$. Finally, probabilities (measured on the y-axis) for scoring function $i$ are calculated as

$$P_i = \frac{1}{N} \sum_{j}^{N} I(d_i^{(j)} < t_j).$$

Changing $\tau$ on the x-axis helps us to understand how close each pruning method performs to the best scoring one through the probabilistic information.

The performance profiles show several important effects:

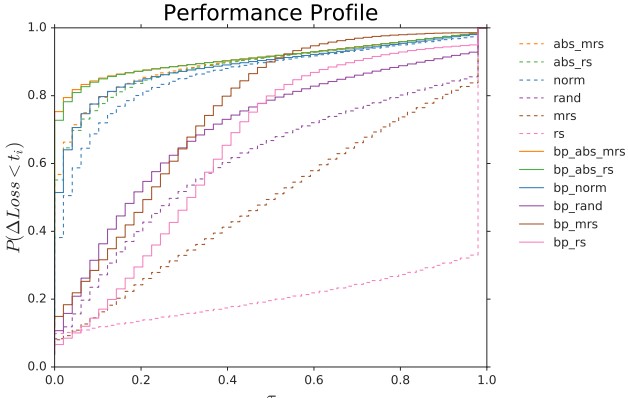

Figure 5: Performance profiles of scoring functions calculated from all experiments we ran for Cifar-10. The y-axis denotes the probability for a particular scoring function to have a pruning penalty smaller than the threshold $t_i = \min(\Delta Loss)_i * \tau + \max(\Delta Loss)_i * (1 - \tau)$ where the $\min$ and $\max$ are calculated separately among the scoring functions for each time step $i$. The x-axis denotes the interpolation constant $\tau$ that determines the exact threshold $t_i$ used for specific pruning measurements. Bias propagation improves the performance of every scoring function considered.

- Using Mean Replacement(lines without dashes) consistently improves performance. This observation agrees with result in the previous section and results provided by Molchanov et al. (2016).

- ABS_MRS and ABS_RS have very similar performance, with the former potentially providing a small improvement over the latter. We have observed a strong overlap between the units selected for pruning by these two methods.

- The direct first order approximations of the pruning penalty, MRS and RS, perform worse than random selection. This is very striking since it shows that the methods using pure first order approximations can have large error terms and cause serious damage to the networks.

To gain insight into this last phenomenon, we plot the output histogram of units pruned with three of our methods in Figure 6b. The corresponding pruning penalties are shown in Figure 6a. Figure 6b reveals the units selected by the absolute valued Taylor approximation have smaller squared outputs

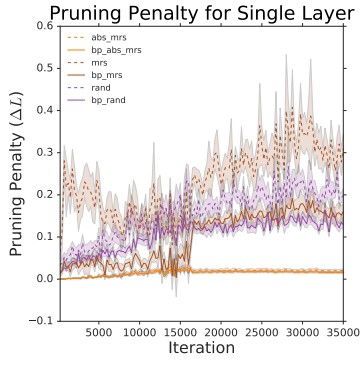
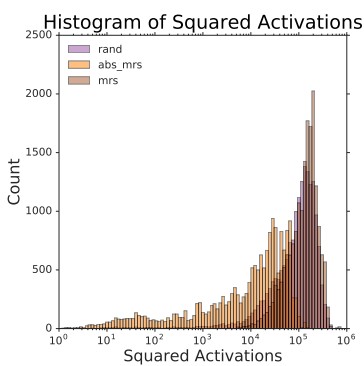

(a) Pruning Penalties through Training    (b) Histogram of Squared Activation's

Figure 6: **Figure 6a** shows the pruning penalties for the specific experiment setting averaged over 8 seeds. We use MEDIUM_CONV network and perform pruning experiments on the second convolutional layer using a pruning fraction of 0.1. **Figure 6b** is the histogram of the squared activation's of the pruned units from the same experiment. The distribution for MRS is includes many samples with high squared norm suggesting a high error term for the approximation.

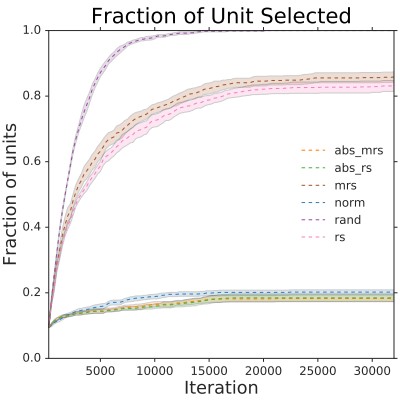
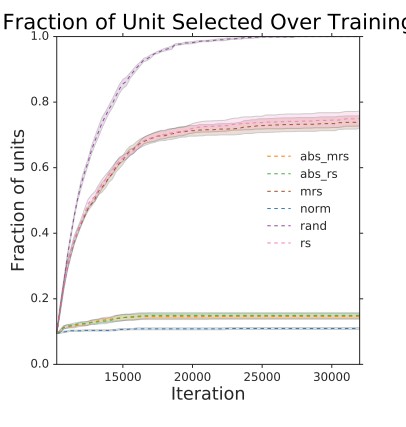

(a) from the start of the training                    (b) after 10k steps

Figure 7: Fraction of units selected at least once by the scoring algorithm accumulated throughout the training. We use the same experimental setting as Figure 6 and show how many different units the scoring algorithm selects throughout the training. Random scorer selects all units at least once by the iteration number 15000. Successful scoring functions somehow stay consistent with their choices and choose a small subset of units in the convolutional layer. In **Figure 7b** we repeat the same plot discarding the measurements taken before step 10000. The set of units chosen by the scoring function decreases later in the training.

and therefore they provide better approximations keeping the error term of the Taylor expansion small(see Appendix 6.1 for further discussion).

### 4.4    ORDERING AMONG UNIT SALIENCIES DOESN'T CHANGE MUCH DURING TRAINING

We now use the same experimental setup as the Figure 6 but keep track of the accumulated set of units pruned at different time steps during training. The curves shown in Figure 7a indicate which fraction of the units of a specific layer have been pruned at least once before the number of iterations specified on the horizontal axis. These curves quickly stop increasing, indicating that the scoring functions quickly select a stable set of units for pruning.

The top curves we see in the performance profile (Figure 5) appear at the bottom in Figure 7. In other words, our best performing pruning methods selects a small subset of units for pruning relatively early during training and keep this set consistent afterwards. This is striking because it indicates that the "winning ticket" discussed by Frankle & Carbin (2018) can be identified relatively early during training.

## 5    DISCUSSION AND FUTURE WORK

This work presents an experimental comparison of unit pruning strategies throughout the training process. We introduce the mean replacement approach and show that it substantially reduces the impact of the unit removal on the loss function. We also show that fine-tuning the pruned networks does not reduce the mean replacement advantage very quickly. We argue that direct first order approximation of the pruning penalty are poor predictors of the pruning penalty incurred by the simultaneous removal of multiple units because the neglected high order terms can become significant. In contrast the absolute value versions of these approximations achieve the best performance. Finally we provide some evidence showing that our best pruning methods identify a stable set of prunable units relatively early in the training process.

This last observation begs for future work. Can we combine pruning and training in a manner that reduces the computational training cost to a quantity comparable to training the "winning ticket" network?

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

Table 2: Experiments for Cifar-10

| hParam | Values |
|---|---|
| Pruning Factor | [1%, 5%, 10%, 20%, 50%, 1 unit] |
| Seed | [0, 1, 2, 3, 4, 5, 6, 7] |
| Model Architecture | [small_conv, medium_conv, vgg_11] |
| Layers Pruned | [all, firstconv, midconv, lastconv, firstdense] |
| Total:1440 | |

Table 3: Experiments for Imagenet-2012

| hParam | Values |
|---|---|
| pruningfactor(if float) or pruningcount(if int) | [0.05, 0.1, 0.3, 1] |
| seed | [0, 1, 2] |
| Model Architecture | [alexnet, vgg11] |
| Layers Pruned | [all, firstconv, midconv, firstdense] |
| Total:96 | |

## 6 Appendix

### 6.1 Mean Replacement Saliency and Absolute Valued Approximations

If we decided that we will be using Mean Replacement as our pruning method, we can define a new scoring function, i.e. the first order Taylor approximation of the pruning penalty after mean replacement. We name this new saliency function as *Mean Replacement Saliency (MRS)* Let us parameterize the loss as a function with activations and write down the first order approximation of the absolute change in the loss.

$$MRS := |L(\bar{\boldsymbol{a}}^l) - L(\boldsymbol{a}^l)| = |\Delta \boldsymbol{a}^l \odot \nabla_{\boldsymbol{a}^l} L(\boldsymbol{a}^l)| + O(||\bar{\boldsymbol{a}}^l||^2) \tag{5}$$

where

$$\Delta \boldsymbol{a}_i = (\bar{\boldsymbol{a}}_i^l - \boldsymbol{a}_i^l) \tag{6}$$

If we were interested in the **average change** in the loss we can write down the Equation 5 without the absolute values. In other words approximations on absolute change penalizes both directions, emphasizing the change in the neural network itself rather then the loss function.

### 6.2 Experimental Details

Pruning can be done at any part of the training. Since we want to make our results as general as possible we perform experiments during the training every 250 or 10000 steps for Cifar-10 and Imagenet-2012 respectively and measure the pruning penalties. Different settings we use in our experiments summarized in Table 2. We perform pruning for different sets of constraints. First we select which layers to prune. This can be a single layer or all layers at once. For pruning single layers we select the first, middle and last convolutional layers and the first dense layer of each network. We use a fraction or count to decide how many units we will be pruning at each measurement step. If we are pruning all layers, we use the same fraction/count for all layers. To be able to compare

our results with Molchanov et al. (2016), we also perform single unit removals. To be able generate confidence intervals, we perform 8 experiments with each setting.

For each combination of settings in Table 2, we pause the training every 250 iteration and perform pruning measurements on the copied model. These measurements include calculation of scoring functions, pruning selected units, optionally doing the bias propagation and finally measuring the pruning penalty. We perform pruning measurements for all scoring functions during the same run creating an exact copy of the model separately with and without mean propagation. This brings us 12 pruning penalty curves for each experiment.

For each experiment we independently sample a fixed validation subset of size 1000 (cifar10) and 10000 (imagenet) from training set. This validation set is used to calculate scoring functions, mean replacement, pruning penalty, and training loss. We set our batch size to 64 for both datasets and perform training for 60 epochs with a learning drop of factor 10 at epoch 45.

## 6.3 MODELS USED

**ALEXNET**
**2DConv**, out_channels=96, filter_size=[11, 11], strides=(4, 4)
**MaxPooling**, pooling_size=3, stride=2
**2DConv**, out_channels=256, filter_size=[5, 5],
**MaxPooling**, pooling_size=3, stride=2
**2DConv**, out_channels=384, filter_size=[3, 3],
**2DConv**, out_channels=384, filter_size=[3, 3],
**2DConv**, out_channels=256, filter_size=[3, 3],
**Flatten**
**Dense**, out_features=4096
**Dense**, out_features=4096

**VGG_11**
**2DConv**, out_channels=64, filter_size=3, padding=same
**MaxPooling**, pooling_size=2, stride=2
**2DConv**, out_channels=128, filter_size=3, padding=same
**MaxPooling**, pooling_size=2, stride=2
**2DConv**, out_channels=256, filter_size=3, padding=same
**2DConv**, out_channels=256, filter_size=3, padding=same
**MaxPooling**, pooling_size=2, stride=2
**2DConv**, out_channels=512, filter_size=3, padding=same
**2DConv**, out_channels=512, filter_size=3, padding=same
**MaxPooling**, pooling_size=2, stride=2
**2DConv**, out_channels=512, filter_size=3, padding=same
**2DConv**, out_channels=512, filter_size=3, padding=same
**MaxPooling**, pooling_size=2, stride=2
**Flatten**
**Dense**, out_features=512
**Dense**, out_features=512

**SMALL_CONV**
**2DConv**, out_channels=32, filter_size=5,
**MaxPooling**, pooling_size=2, stride=2
**2DConv**, out_channels=64, filter_size=3,
**MaxPooling**, pooling_size=2, stride=2
**2DConv**, out_channels=128, filter_size=3,
**MaxPooling**, pooling_size=2, stride=2
**Flatten**
**Dense**, out_features=512
**Dense**, out_features=128

**MEDIUM_CONV**
**2DConv**, out_channels=64, filter_size=5,

**MaxPooling**, pooling_size=2, stride=2
**2DConv**, out_channels=128, filter_size=3,
**MaxPooling**, pooling_size=2, stride=2
**2DConv**, out_channels=256, filter_size=3,
**MaxPooling**, pooling_size=2, stride=2
**Flatten**
**Dense**, out_features=1024
**Dense**, out_features=256

## 6.4 Scatter Plots for Disjoint Subsets of the Cifar10 Experiments

One down side of having a single plot aggregating all experiments is that anomalies in some small subset of the experiments might get shadowed by the rest of the experiments. Even though it is not feasible to share every single plot without any aggregation, in this section we like to split our experiments into 5 disjoint subsets and plot them separately. Figure 8 shows these 5 disjoint subset of experiments. Pruning single layers (Figure 8 b-d), we observe that the gains seem like more prominent(greater decrease in pruning penalty) compare to the other layers.

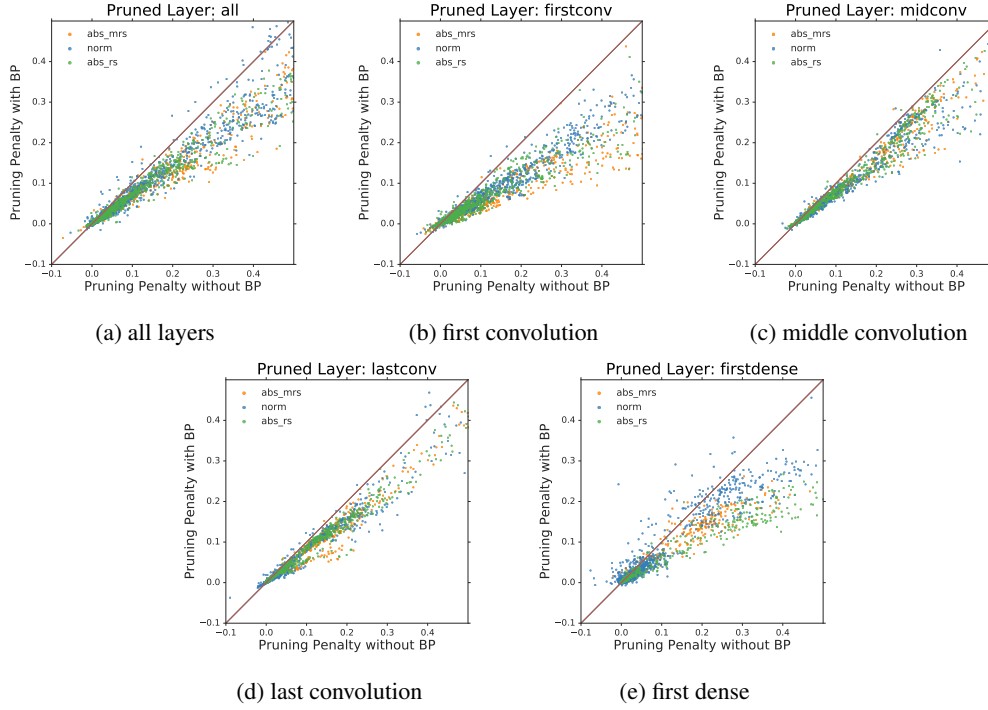

Figure 8: Scatter plots generated similar to the ones in Section 4.1. However the entire set of results are partitioned according to the layer/s pruned.

## 6.5 Additional Scatter Plots with Fine-tuning Steps

In Section 4.2 we argued that the mean replacement helps optimization by reducing the gap between loss before and after pruning. Particularly, we focused on the measurements done in the second half of the training. In this section we like to share complimentary data, where instead of the second half of the training (where the networks are mostly converged), we plot the first half in Figure 9. As expected we see more points in the negative regime, where the final loss is smaller than the loss before pruning due to the fine-tuning steps taken after pruning.

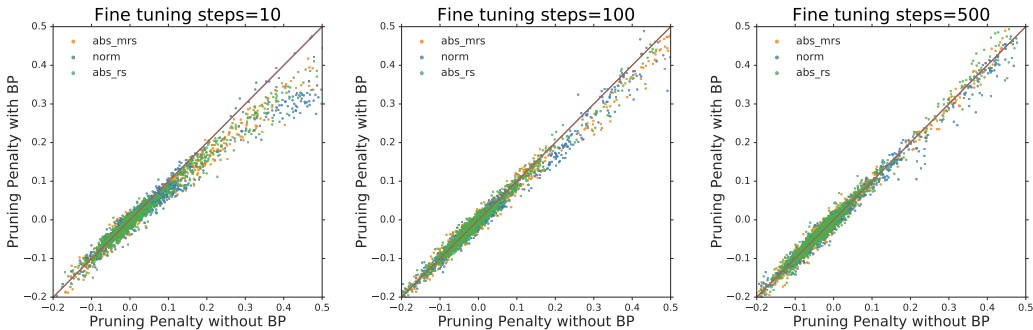

Figure 9: Pruning penalties measured after $N$ fine tuning steps. For fine tuning steps a batch size of 64 and a learning rate of 1e-3 are used. The data is gathered from the first half of Cifar-10 experiments.

The effectiveness of our method diminishes with increased number of fine tuning steps compare to the plots shared in Section 4.2. However we think that this comparison might not tell us a lot, due to the ongoing optimization problem and its inference with the effect of mean replacement.

## 6.6 ITERATIVE PRUNING EXPERIMENTS

As our work and experiments focus on minimizing the immediate damage on the network , many practical applications allow computational budget required for iterative pruning and fine-tuning. Results we got in Section 4.2 and Appendix 6.5 suggests that, a network with the same sparsity but slightly worst starting point (no bias propagation) would possibly catch up the one with better starting point (mean replaced version). In this section, we like to extend our investigation one step further and perform iterative pruning experiments with extended number of fine-tuning steps to answer whether this two starting points have different optimization paths leading to two different end points.

As number of units pruned in one pass approaches to the total number of units, all pruning methods approaches to the random scoring function. To minimize this effect we employ iterative pruning strategy, where we prune 1% of a layer at a time and perform 100 fine tuning steps in between until the target pruning fraction is reached. We prune all layers in the MEDIUM_CONV together starting from iteration 60000. We perform 93750 iterations(120 epoch) in total and report the average values for training loss, test loss and test accuracy over 8 different runs with 80% confidence intervals in Figure 10. To our surprise, results in Figure 10 suggests various pruning methods perform slightly

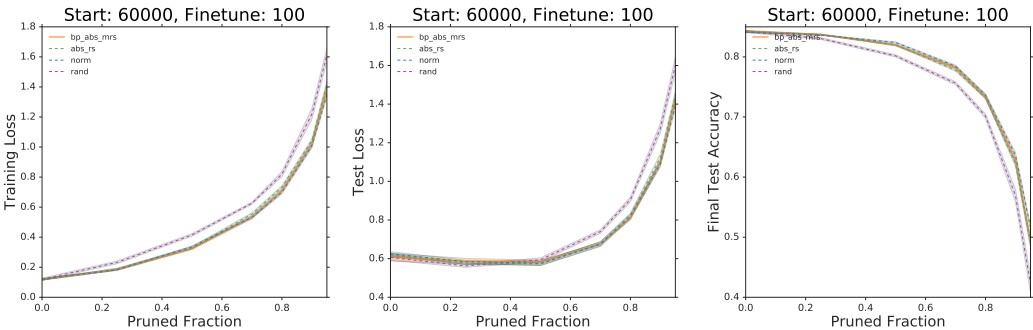

Figure 10: Performance of various pruning methods in iterative pruning setting. Networks are pruned iteratively starting from epoch 60000. We prune 1% of each layer and fine-tune with learning rate of 1e-3 for 100 iterations in between, until pruning targets are reached.

better than random in our experimental setting. We can also see the regularization effect of the pruning (test loss increases much slower than the training loss with increased target sparsity). To

investigate these results further we repeat the same experiment with VGG_11 network, 10 fine-tuning steps between pruning iterations and a starting iteration of 10000 (changing one at a time, total 7 new set of experiments). Results from these experiments confirm the picture in Figure 10 and sometimes we observe, curves running even closer to each other.

As a sanity check we use the data from Figure 4 and group the pruning penalties to compare different pruning methods. We make 2 comparisons in one plot by changing the axis that represents our best performing method **bp_abs_mrs** keeping the two cloud of points on different sides of the diagonal. By generating these 1-1 comparisons for increasing number of fine-tuning steps, we observe how the comparisons evolve. If all methods would perform exactly the same, we would expect all points to converge to the diagonal and indeed, we observe such a movement in Figure 11 when we increase the number of fine tuning steps. Blue cloud of points become almost perfectly diagonal, whereas the orange cloud(comparison against random scoring function) employs a rather slower movement supporting the picture we observe in Figure 10, particularly the difference between random and other methdods.

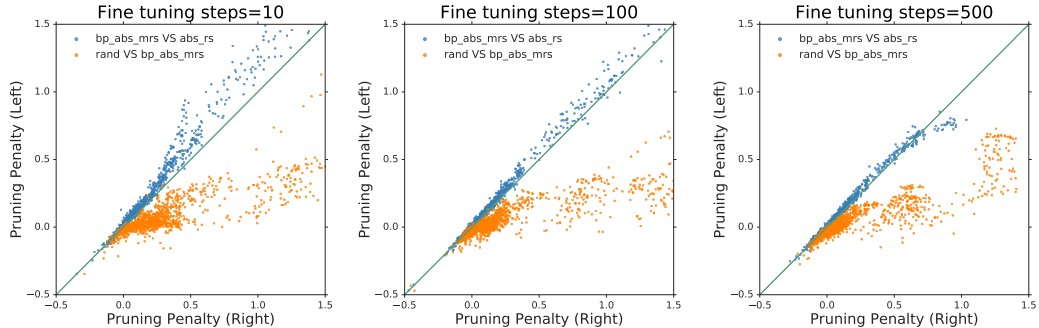

Figure 11: Pruning penalties from the experiments in Section 4.2 grouped by various pruning methods. We do 2 comparisons in 1 graph by using different colours. For a label 'Y VS X' x-axis represents pruning penalties when pruning method X is used and y-axis represents the pruning penalties when the method Y is used. Difference among different pruning methods diminishes with increased number of fine tuning steps.

In a future work, we plan to continue investigating the discrepancy between our findings in this section with experiments made by Molchanov et al. (2016) and Luo et al. (2017). We suspect that the exact pruning strategy used, along with the hyper-parameters like learning-rate may have an import effect on the performance of pruning methods and explain the different results we observed.

