# OpenReview forum: "Mean Replacement Pruning  "
_ICLR.cc/2019/Conference_

### Official Review · AnonReviewer2 · 2018-11-03
**Interesting and simple method, but needs clarification w.r.t. related methods and results**

**Rating:** 5
**Confidence:** 3

**Review:**

This paper proposes a simple improvement to methods for unit pruning. After identifying a unit to remove (selected by the experimenter’s pruning heuristic of choice), the activation of that unit is approximately incorporated into the subsequent unit by “mean replacement”. The mean unit activation (computed on a small subset of the training set) is multiplied by each outgoing weight (or convolutional filter) and added to each corresponding bias instead. Experiments show this method is generally better than the typical method of zero-replacement before fine-tuning, though the advantage is smaller after several epochs of fine-tuning.

While I find this paper intriguing and applaud the extensive experimentation and documentation, I have some concerns as well:
	1. There are unanswered questions about how this method relates to existing work. It is not clear from the paper how the “mean replacement” method differs from the two most related works (Ye, 2018) and (Morcos, 2018), which propose variations on replacing units with constant values or mean activations, respectively. Also, why does the method in this paper seem to yield good results, while the related method (Morcos, 2018) yields “inferior performance”?
	2. The results are stated to only apply to networks “without batch normalization”. The reason seems intuitive: any change that can be merely rolled into the bias will be lost after normalization (depending perhaps on the ordering of normalization and the non-linearities). This leaves an annually decreasing fraction of networks to which this method is applicable, given the widespread use of batch norm.
	3. Critically, it’s difficult to compare this work against other pruning works given the lack of results reported in terms of final test error and the lack of the ubiquitous “error vs. %-pruned” plot.

Overall, this paper is lacking some clarity, may be limited in originality, may be helpful for some common networks and composable with other pruning methods (significance), but has a good quality evaluation (subject to the clarity issues). I’m rating this paper below the threshold given the limitations, but I’m willing to consider an upgrade to the score if these questions are addressed.

Other notes:
	4. What is your definition of a convolutional “pruning unit”? (From context, I’d presume it corresponds to an output activation map.)
	5. In Section 3.1:  replace “in practice, people …” with  something like “in practice, it is common to”.
	6. In Equation 3, is the absolute value of the pruning penalty used in the evaluation?
	7. In the footnote in Section 3.2, how many training samples are needed for a good approximation? How many are used in the experiments?
	8. There are a couple typos in Section 3.2: “replacing -the- these units with zeroes” and “each of these output*s*”.
	9. Presumably the “\Delta Loss after pruning” in Figures 2-6 is validation or test loss, not training loss? Is this the cross-entropy loss? Also, it would be much easier to compare to other papers if test accuracy were reported instead or in addition.
	10. In Figure 4, the cost to recover using fine-tuning seems to be only roughly 2% of the original training time. How much time is lost to the process of computing the average unit activation?

UPDATE: I've raised the score slightly to 5 after the rebuttals and revisions.

---

> ### Author Response · Authors · 2018-11-23
> **Thanks for your extensive review and comments.**
>
> Thank you very much for your thorough comments and valuable suggestions. We hope the responses/clarifications below would help.
>
> (1) Ye et al. (2018), proposes a pruning method that penalizes the variance of the output distribution of a unit. When units have low variance (generating almost constant values), they remove the unit by propagating the mean values to the next layer. In our method we show that such a trick can be utilized in other pruning scenarios that don’t involve batch normalization or variance regularization. In particular, we show that mean replacement is the optimal update minimizing the reconstruction error on the next layer and empirically show that it does indeed reduce the pruning penalty in a variety of settings. Morcos et al. (2018) briefly mentions the possibility of using mean replacement in the ablation setting. In the single result shared in Section A1, they prune/ablate the last layer of their convolutional network, since that would be the only layer without batch normalization.
>
> (2) We agree that there is a strong trend with using batch normalization in neural network training. However there is no guarantee that this trend will last or/and become a standard. There are also cases where batch normalization is not practical/preferred, like small batch training (due to memory limitations of big networks) and RNN training. We would also like to point out that our method is applicable in networks with instance or layer normalization.
>
> (3) We updated the table of scoring functions in Section 3.4 with related references. We hope that it will guide the reader to understand differences between our work and previous work. Following the common feedback we got from the reviewers, we performed some additional experiments. In our experiments on Cifar-10, we didn't find any significant difference among different pruning methods despite the reported differences in others works. We share this surprising initial results in Section 6.6 along with a discussion about possible explanations.
>
> (4) We use the same notation as Morcos et al. 2018 in our work. Given a convolutional kernel W with dimensions (input channels, H, W, output channels), a unit i corresponds to the parameters W[:,:,:,i]. In the case of a dense layer with parameters W of size (input channels, output channels) it corresponds to W[:,i].
> (5) Updated.
> (6) We used the average change in the loss to evaluate the performance. We expect the two measures(absolute and average change in the loss) to be very similar, especially when the pruning fraction is high.
> (7) We forgot to share these numbers, thanks for the note. It is 1000 and 10000 for cifar-10 and imagenet.  We updated Section 3.4 to include these numbers.
> (8) Thanks for the feedback, we updated the part and had a second pass on the whole paper.
> (9) In Figures 2-6 we report pruning penalties (cross entropy loss) using a set of 1000 (Cifar-10) and 10000 (Imagenet2012) samples from the training set. Our motivation is that reducing the pruning penalty (training loss) would help the optimization by both/either reducing number of fine tuning steps needed and/or improving final performance. Our results support the first and undermine the second.
> (10) In practice we  would suggest using a running average of the mean outputs, which would require constant number of FLOPS per sample (linear in number of units in the network). Since our initial set of experiments don't have end-2-end pruning experiments we haven't implemented such a feature and measure its effectiveness.

---

> > ### Comment · AnonReviewer2 · 2018-12-05
> > **Thanks and a couple more questions**
> >
> > Thanks for your detailed response. Regarding (1), it seems that a reasonable summary would be that the mean replacement idea is described in both (Ye,2018) and (Morcos,2018), albeit in conjunction with other methods in both cases. So, the contribution of this paper is thoroughly experimenting on mean replacement in isolation. Is that fair?
> >
> > Regarding (3), thanks for providing some results with test accuracy. Fig. 10 shows on VGG-11 (with which dataset?) that mean replacement makes little/no difference with 100 fine tuning steps between pruning iterations. But presumably, based on the other plots like Figs. 4 and 11, test error is worse without mean replacement at lower levels of fine-tuning. You show this for training error, but do you have any results in the paper indicating this effect for test error?

---

> > > ### Author Response · Authors · 2018-12-06
> > > **Further Clarifications**
> > >
> > > (1) Yes, it can be fair to say so. Propagating constant values to the next layer (Ye, 2018) and ablating units with their mean values (Morcos, 2018) were used/mentioned in recent work in different contexts. In addition to running a wide set of experiments, our work shows that replacing units with the mean activation value is the optimal bias update that minimizes the reconstruction loss in the next layer. We also introduce a new scoring function (i.e the first order approximation of the absolute change in the loss when mean replacement pruning is used).
> > >
> > > (4) In Figure 10 (networks trained on Cifar-10), we report measurements taken at the end of training. In other words, there are almost 30k fine-tuning steps after pruning. In Figs. 4 and 11 the pruning penalty is calculated after a small number of fine tuning steps (10-500). These initial results (Fig. 10) suggest that mean replacement reduces the number of fine-tuning steps needed, however the network converges to the same energy level if trained long enough.
> > >
> > > When we generate Fig. 4 with test loss, we see a similar picture (almost all points are under diagonal).

---

> > > > ### Comment · AnonReviewer2 · 2018-12-10
> > > > **Follow-up**
> > > >
> > > > Thanks for the additional response. Overall, I'm somewhat conflicted on this paper. The revisions have made the paper stronger. I generally like the thorough experiments, and, were the idea to be entirely novel, the scope of the analysis and experiments would be reasonably compelling.
> > > >
> > > > But (1) the basic idea is already known in a couple of publications. And regardless of novelty, (2) the significance is limited: (a) the method seems applicable only to networks without batch normalization, and (b) the initial advantage of pruning with this method is matched by instead using some number of fine-tuning steps (though number of additional steps isn't estimated in the paper). As such, I'm not sure who really needs to know about this method or what follow-up work it could inspire. Unfortunately, while the paper may be a no-brainer to accept at a workshop, but I don't think it meets the bar for an ICLR conference publication.

---

> > > > > ### Author Response · Authors · 2018-12-11
> > > > > **Fine-tuning**
> > > > >
> > > > > Thank you for your comment, although we believe one should not underestimate the impact that reducing the number of required fine-tuning steps might have.
> > > > >
> > > > > Indeed, in the case of incremental pruning, fine-tuning steps can take a significant portion of the total training time and reducing them is desirable.
> > > > >
> > > > > Another result worth emphasizing once more is the fact that, based on the metric chosen, pruning methods can be found to be nearly equivalent. In a field where there are many new methods proposed, we believe it is important to show the sensitivity of the result to the particular metric of choice, which emphasizes the importance of defining ahead of time what the goal of pruning is.

---

### Official Review · AnonReviewer3 · 2018-11-04
**Overall score: 4**

**Rating:** 4
**Confidence:** 4

**Review:**

1. Pruning neurons in pre-trained CNNs is a very important issue in deep learning, and there are a number of related works have been investigated in Section 2. However, it is very strange that, I did not see any comparison experiments to these related works in this paper.

2. The presentation of the experiment part is also wired, to report compression rates, speed-up rates, and accuracy might have a more explicit demonstration.

3. ''This is often done by replacing the these units with zeroes". However, in previous works, we can directly establish a compact network with fewer neurons after pruning some unimportant neurons. Thus, some considerations and motivations in Section 3.2 seem wrong.

4. It seems that the neural network after using the proposed method has the same architecture as that of the original network, but some of it neurons are represented as mean replacement. Therefore, the compression and speed-up rates of the proposed method would be hard to implement in practice.

5. The paper should be further proofread. There are considerable grammar mistakes and unclear sentences.

---

> ### Author Response · Authors · 2018-11-23
> **Updates and Clarifications**
>
> Thank you for your review.
>
> (1) We updated the table of scoring functions in Section 3.4 with related references. We hope that it will guide the reader to understand differences between our work and previous work. We also updated the last paragraph of Section 2 to highlight the difference between our work and Ye et al, (2018), Morcos et al., (2018).
>
> (2) Even though pruning is widely used for model compression, recent work highlights a promising direction where pruning is used during training to reduce training time and improve final performance (Han et al. (2016), Frankle & Carbin (2018)). Therefore we focused on pruning experiments sampled from the entire length of a training job. Our motivation is that reducing the pruning penalty helps optimization by reducing the number of fine tuning steps needed for reaching the same level. However, following the common feedback we got from the reviewers, we performed some additional experiments. In our experimental setting, we didn't find any significant difference among different pruning methods. We share this surprising results in Section 6.6 along with a discussion about possible explanations.
>
> (3) Replacing a unit with zeros indeed corresponds to removing it along with all the outgoing weights (it represents the same function). In our experiments we use masking to emulate this behaviour, however as you suggest, one can establish a smaller network in practice immediately if needed.
>
> (4) Our method replaces units with their mean output values. In practice, constant values are propagated to the next layer and units are removed as normal (see Figure 1). Mean output values can be aggregated during the training in an online fashion and bias propagation is a very cheap operation (single matrix multiplication). Therefore, it is a practical method.
>
> (5) We are sorry about the mistakes slipped and we did further proofread the paper.
>
> Han, S., Pool, J., Narang, S., Mao, H., Gong, E., Tang, S., … Dally, W. J. (2016). DSD: Dense-Sparse-Dense Training for Deep Neural Networks. Retrieved from http://arxiv.org/abs/1607.04381

---

### Official Review · AnonReviewer1 · 2018-11-05
**Some clarity issues**

**Rating:** 5
**Confidence:** 3

**Review:**

This paper presents a mean-replacement pruning strategy and utilizes the absolute-valued Taylor expansion as the scoring function for the pruning. Some computer vision problems are used as test beds to empirically show the effect of the employment of bias-propagation and different scoring functions. The empirical results validates the effectiveness of bias-propagation and absolute-valued Taylor expansion scoring functions.

The work is generally well-written and the results are promising, and the theoretical explanation in 3.3 is intriguing. However, I think the following issues need some further clarifications:
1. What's the exact difference and connection between the mean-replacement pruning technique, and the bias-propagation technique in Ye et al., (2018) and the mean activation technique in Morcos et al. (2018)? The authors only mention that mean replacement pruning extends the idea in Ye et al. (2018) to the non-constrained training setting, but it is very unclear what "constraints" are talked about. Some more detailed and formal comparisons should be added, together with potential empirical comparisons.
2. In the abstract, the authors claim that they "adapt an existing score function ...", but from the main text it seems that absolute-valued Taylor expansion score is exactly the same one in Molchanov et al. (2016). Is this a typo (or misleading claim) in the abstract?
3. There are no comparisons of the approach proposed in this paper with some existing state-of-the-art, apart from some simple comparisons between whether bias-propagation is adopted and some inner comparisons among different scoring functions.

It would also be much better if some charts/tables with certain metrics for improvement apart from pruning penalties (e.g., compression rates, or inference speed, etc.) instead of simply illustrative figures are shown.

### some smaller suggestions/typos ###
1. The plot legends/labels are kind of inconsistent with the description before the figures. For example, in the main text the authors mainly use "pruning penalty", while in the figures the y-axes are typically labelled as "\Delta-loss after pruning", and the plot tag at page 5 bottom is different from those used in the plots, which introduces some unnecessary confusion.
2. It is very unclear how the authors arrive at the conclusion "This results suggests ... the training process" from the "winning ticket" hypothesis.
3. Several typos that can be easily spell-checked (e.g., "the effect or pruning" -> "the effect of pruning", etc.).

I hope the authors can address these issues. Thanks!

---

> ### Author Response · Authors · 2018-11-23
> **Framing our work and contributions better**
>
> Thank you for your review and valuable comments. We would like to provide clarifications/updates about the questions raised.
>
> (1) Our work extends the setting in Ye et al, (2018) (networks with batch normalization and units with very small variance) to networks without batch-normalization, un-regularized training and shows that replacing a unit with its mean value is a better at reducing the immediate damage compared to mere removal.  Morcos et al., (2018) compare mere removal with mean replacement in the context of ablation studies using a network with batch normalization. Since they use layers with batch normalization they are able to mean replace the final layer of the network only (Batch normalization should remove any constant signal coming from the previous layer and therefore we shouldn't see any difference between the two methods in their setting, except the output layer of the network.). Our experiments involve a much greater variety of settings and show that Mean Replacement indeed works better in general.
>
> (2) Our scoring function and Molchanov et al. (2016) are first order approximations of two different values. We updated the table of scoring functions in Section 3.4 with related references. We hope that it will guide the reader to understand differences between our work and previous work.
>
> (3) We intentionally avoided running end to end pruning experiments in our work. Our assumption was that reducing the change in the training loss should help any pruning strategy that employs any of the saliency scores used. However, we agree that this connection is not clear and needs further investigation. Therefore we  ran additional experiments where we perform iterative pruning with fix training budget. In our experiments, surprisingly, we didn't find any significant difference among `non-random` methods despite the reported differences in others works. Even though the results don't support our case, we like to share these initial results in the appendix (Section 6.6) with a discussion on possible reasons.
>
> (3) And finally regarding minor suggestions, (1) we updated with 'Pruning Penalty` following your suggestion (2) Detecting lottery ticket early in the training is important, since one could reduce the size of the network and the training time. This result, if general enough, promises a whole new direction for pruning research. We updated our introduction indicating this point. (3) spell-checked.

---

> > ### Comment · AnonReviewer1 · 2018-12-07
> > **Thanks for the response!**
> >
> > I appreciate the authors' detailed responses and the modifications in the new draft. In particular, table 1 in the new draft makes the legends in the plots much clearer. However, I still have two concerns:
> >
> > 1. regarding the "winning ticket" hypothesis, do you want to emphasize that the pruning can be done after a short pre-training of the large network, followed by re-training from the same initialization of the pruned network, which would result in both faster training and smaller network size (and faster inference)?
> >
> > 2. Another thing is about the novelty and significance of the work, and its relationship to other methods. I would suggest the authors to add an algorithm frame describing the prototype of a general pruning approach in neural networks, with consistent mathematical notations about scoring functions, approximated penalty, and the mathematical formulation of mean replacing/back-propagation, and how they interact and combine with the other components. The examples of different scoring functions and approximated penalties can then be mathematically listed as special cases of the abstractions in the algorithm frame. For now, although the draft has been largely improved to show difference and connections between different methods, for a non-expert about neural network pruning like me, it is still very unclear how different the methodologies are and how the performance metrics are exactly (mathematically) defined. This also prevents me from accurately judging the difference and connection between the current work and the existing literature, as well as understanding the performance of different approaches given the diversity of experimental settings presented in the draft. This is especially the case given some remaining consistency in the notations, for example, what is the loss degradation \Delta L? Is it just the pruning penalty? Similarly, what do \nabla_a L(a) and \Delta a stand for? And how does the i sampled from D_s come into play with them (as i does not even show up)? Also, how is the approximated penalty involved in the entire pruning approach? And I believe that all these can be largely solved with a general algorithm frame added.
> >
> > But anyway, thanks for the responses and updates on the draft!

---

> > > ### Author Response · Authors · 2018-12-07
> > > **General framework**
> > >
> > > Thank you for your comment. We actually considered casting all existing methods as special cases of a general framework but we felt the added layer of abstraction might be confusing to those already familiar with existing methods.
> > >
> > > However, should the paper be accepted and the consensus among reviewers be that this would improve clarity, it would be straightforward for us to include such a framework in the final version.
> > >
> > > Once again, thank you for taking the time to reply to our updates.

---

### Meta-Review · Area_Chair1 · 2018-12-13
**Good writing and experiments, but limited novelty and applicability**

**Confidence:** 5
**Recommendation:** Reject

**Metareview:**

This paper proposes an approach to pruning units in a deep neural network while training is in progress. The idea is to (1) use a specific "scoring function" (the absolute-valued Taylor expansion of the loss) to identify the best units to prune, (2) computing the mean activations of the units to be pruned on a small sample of training data, (3) adding the mean activations multiplied by the outgoing weights into the biases of the next layer's units, and (4) removing the pruned units from the network. Extensive experiments show that this approach to pruning does less immediate damage than the more common zero-replacement approach, that this advantage remains (but is much smaller) after fine-tuning, and that the importance of units tends not to change much during training. The reviewers liked the quality of the writing and the extensive experimentation, but even after discussion and revision had concerns about the limited novelty of the approach, the fact that the proposed approach is incompatible with batch normalization (which severely limits the range of architectures to which the method may be applied), and were concerned that the proposed method has limited impact after fine-tuning.